# ISOTOPE: ISOform-guided prediction of epiTOPEs in cancer

Juan L. Trincado[1], Marina Reixachs-Solé[2,3], Judith Pérez-Granado[4], Tim Fugmann[5¤a], Ferran Sanz[4], Jun Yokota[6], Eduardo Eyras[2,3,7,8]*

**1** Josep Carreras Leukemia Research Institute, Badalona, Spain, **2** Australian National University, Canberra, Australia, **3** EMBL Australia Partner Laboratory Network at the Australian National University, Canberra, Australia, **4** Research Programme on Biomedical Informatics (GRIB), Hospital del Mar Medical Research Institute (IMIM), Dept. of Experimental and Health Sciences, Pompeu Fabra University (UPF), Barcelona, Spain, **5** Philochem AG, Otelfingen, Switzerland, **6** National Cancer Center Research Institute (NCCRI), Tokyo, Japan, **7** Catalan Institution for Research and Advanced Studies, Barcelona, Spain, **8** Hospital del Mar Medical Research Institute (IMIM), Barcelona, Spain

¤a Current address: Institute of Biochemistry, Justus Liebig University Giessen (JLU), Giessen, Germany
* eduardo.eyras@anu.edu.au

**Data Availability Statement:** All relevant data are within the manuscript and its Supporting information files.

## Abstract

Immunotherapies provide effective treatments for previously untreatable tumors and identifying tumor-specific epitopes can help elucidate the molecular determinants of therapy response. Here, we describe a pipeline, ISOTOPE (ISOform-guided prediction of epiTOPEs In Cancer), for the comprehensive identification of tumor-specific splicing-derived epitopes. Using RNA sequencing and mass spectrometry for MHC-I associated proteins, ISOTOPE identified neoepitopes from tumor-specific splicing events that are potentially presented by MHC-I complexes. Analysis of multiple samples indicates that splicing alterations may affect the production of self-epitopes and generate more candidate neoepitopes than somatic mutations. Although there was no difference in the number of splicing-derived neoepitopes between responders and non-responders to immune therapy, higher MHC-I binding affinity was associated with a positive response. Our analyses highlight the diversity of the immunogenic impacts of tumor-specific splicing alterations and the importance of studying splicing alterations to fully characterize tumors in the context of immunotherapies. ISOTOPE is available at https://github.com/comprna/ISOTOPE.

## Author summary

Immune cells have the ability to attack tumor cells upon the identification of tumor-specific peptides, i.e., epitopes, that are presented by the major histocompatibility complex (MHC). New cancer immunotherapies that help trigger this process provide a promising therapeutic strategy. One crucial aspect for their success is the ability to determine the molecular properties of a tumor that are informative about the effectiveness of the therapy. Alterations in the way genes are processed to express RNA molecules could lead to the production of new peptides, with some of them potentially being presented as tumor epitopes and facilitate the attack of immune cells. It is therefore essential to facilitate the

**Funding:** This work was supported by the Agencia Estatal de Investigación (AEI), Spanish Government and European Regional Development Fund (FEDER) with grant BIO2017-85364-R (F.S., E.E.), by the Agència de Gestió d'Ajuts Universitaris i de Recerca (AGAUR, Generalitat de Catalunya) with grants SGR2017-1020 (E.E) and 2017 SGR 00519 (F.S), by the Instituto de Salud Carlos III (ISCIII and FEDER) with grants FI18/00034 (J.P-G) and PT17/0009/0014 (F.S), and by the AEI with CEX2018-000782-M (F.S). The Research Programme on Biomedical Informatics (GRIB) is a member of the Spanish National Bioinformatics Institute (INB) supported by ISCIII and FEDER (PT17/0009/0014). The DCEXS is a 'Unidad de Excelencia María de Maeztu' supported by the AEI (CEX2018-000782-M). The funders had no role in study design, data collection and analysis, decision to publish, or preparation of the manuscript.

**Competing interests:** The authors have declared that no competing interests exist.

identification of these splicing-derived epitopes. In this work, we describe a computational pipeline that performs a comprehensive identification of splicing alterations in a tumor and the potential epitopes that they would produce. Analysis of tumor samples with our pipeline show that responders and non-responders to immune therapy do not show differences in the number of splicing-derived epitopes, but splicing neoepitopes have higher affinity to the MHC complex in responders. Our new pipeline facilitates the genome-scale analysis of the role of splicing alterations in shaping the molecular properties that influence response to immunotherapy.

## Introduction

Recent developments in the modulation of the immune system have revolutionized the clinical management of previously untreatable tumors. In particular, therapies targeting negative regulators of immune response, i.e. immune checkpoint inhibitors, have shown prolonged remission in several tumor types [1]. However, these therapies appear to be effective only for about one third of the patients [2]. Thus, characterizing the molecular features driving response to immune therapies is crucial to prospectively identify patients who are most likely to benefit from these agents and avoid exposing resistant patients to unnecessary and potentially harmful treatments.

The ability of the T-cells infiltrating the tumor tissue to identify and attack malignant cells relies on tumor cells maintaining sufficient antigenicity. An approach to estimate the antigenicity of a tumor is through the calculation of the frequency of somatic mutations as a proxy for the abundance of tumor neoantigens. This has led to the identification of an association between response to checkpoint inhibitors and tumor mutation burden (TMB) in tumors such as melanoma [3,4], urothelial carcinoma [5], and lung cancer [2,6]. Furthermore, analysis of how somatic substitutions and indels impact the protein products in tumor cells has enabled the identification of cancer-specific neoepitopes [7,8] that can trigger the attack of the immune system against tumor cells during treatment with immune checkpoint inhibitors. However, TMB or mutation-derived neoepitopes can only explain a fraction of the responders [9], and hence, other molecular signatures and sources of neoepitopes need be identified.

Recently, tumor-specific transcriptome alterations have been shown to be a source of neoantigens that can be presented by the MHC complexes and recognized by T-cells. These include gene fusions [10], RNA editing [11], cryptic expression [12,13], and tumor-specific splicing [12,14–16]. In particular, the aberrant selection of splice sites and exon-exon junctions [14,15] or the retention of introns [12,16] in tumors represents an additional potential source of cancer neoepitopes. However, it is not clear yet whether these splicing-derived neoepitopes provide a mechanism to elicit cancer-specific immune responses and whether they may improve patient response to immune therapies.

To address these questions and expand the analysis of splicing-derived neoepitopes in cancer, we developed ISOTOPE (ISOform-guided prediction of epiTOPEs in cancer), a pipeline to exhaustively identify the immunogenic impacts from tumor-specific splicing alterations. ISOTOPE identifies splicing alterations that are specific to each tumor sample in comparison with a comprehensive set of normal samples and calculates the impact on the encoded proteins and the candidate neoepitopes. ISOTOPE also calculates native epitopes that are not present in the altered isoform, i.e., splicing-affected self-epitopes. Our analyses provide evidence that splicing alterations can modify the repertoire of epitopes in tumors and potentially impact the

response to immune therapy. ISOTOPE facilitates the study of splicing alterations to fully characterize the determinants of response to immunotherapies.

## Results

### Comprehensive identification of tumor-specific splicing-derived epitopes

ISOTOPE identifies tumor-specific splicing alterations by generating a catalogue of all exon-exon junctions calculated from RNA sequencing (RNA-seq) reads from each individual tumor sample, filtering out those that appear in any of the samples from a comprehensive set of normal controls (Fig 1). The remaining junctions are classified into one of four possible types: *de novo* exonization, new exon skipping event, alternative splice site, and intron retention. ISO-TOPE performs an empirical test to establish the significance of each candidate splicing alteration taking into account the read support of the event and the coverage and splicing variation in the same gene locus. This test ensures the robustness of the events detected. Changes in the protein products are predicted through the impact of the splicing alterations on the open reading frames (ORF) of the reference transcriptome. This reference transcriptome is obtained by

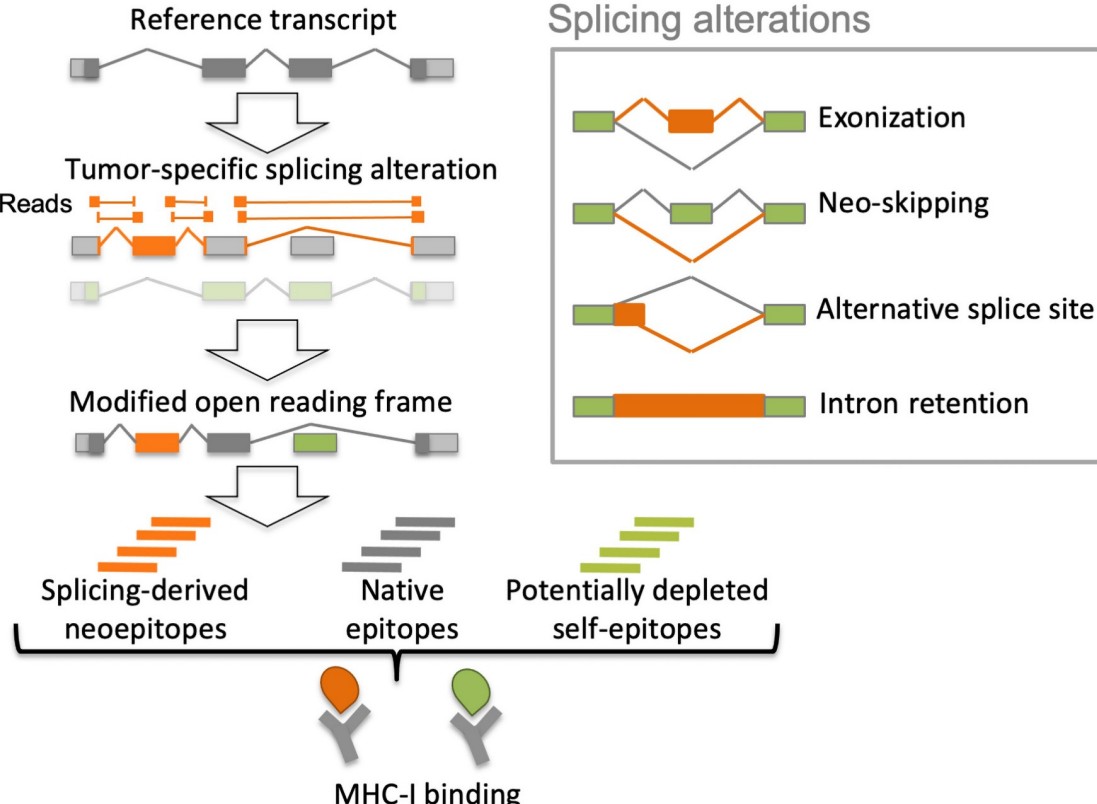

**Fig 1. ISOTOPE pipeline.** Tumor-specific splicing alterations are defined as significant variations with respect to the exon-intron structures expressed in normal samples and are classified into four different types: *de novo* exonization, new exon skipping (neoskipping), alternative (5'/3') splice site, and intron retention. ISOTOPE calculates the modified open reading frame (ORF) from the reference ORF using the splicing alterations, and identifies the candidate splicing-derived neoepitopes and self-epitopes encoded by the reference transcript that would not be present in the modified ORF as a consequence of the splicing alteration. These candidate peptides are then tested for affinity with the MHC complexes (see Methods for details).

selecting from each gene the transcript with the highest mean expression in the control normal samples. ISOTOPE identifies potential epitopes by calculating the binding affinity of the encoded peptides to the major histocompatibility complex class I (MHC-I) or II (MHC-II) using NetMHC-4.0.0 [17]. When the human leukocyte antigen (HLA) type for a patient is not available, this is calculated directly from the tumor RNA-seq sample. ISOTOPE defines candidate splicing-derived neoepitopes as MHC binders that are expressed in the tumor sample but not in the control normal samples, i.e., *splicing-neoepitopes*. Additionally, MHC binders that are expressed in the control sample but are potentially removed by the change in the ORF through the tumor-specific splicing alteration are also calculated and referred to as splicing-affected self-epitopes, *self-epitopes* for short. Further details are provided in the Methods section.

## Detection of cancer-specific splicing-derived epitopes in MHC-I mass-spectrometry

ISOTOPE operates on individual tumor samples, without necessarily having a matched normal sample. We thus first tested the accuracy of predicting HLA types directly from the tumor RNA-seq. We calculated HLA-I and HLA-II types from RNA-seq reads from tumor and matched normal samples for 24 small cell lung cancer (SCLC) patient samples [18] using PHLAT [19] and Seq2HLA [20]. Both methods showed an overall agreement between the HLA predictions from the tumor and the normal RNA-seq data (Fig 2A). However, PHLAT showed greater consistency across most of the HLA types and recovered above 80% of cases for HLA-I and between 65% and 90% for HLA-II types (S1 Table). We thus decided to use PHLAT for further analyses with ISOTOPE.

To test the ability of ISOTOPE to identify potential neoepitopes, we analyzed RNA-seq data and MHC-I associated proteomics data for the cancer cell lines CA46, HL-60 and THP-I [21,22]. *De novo* exonization was the least common of all splicing event types, whereas new junctions skipping one or more exons, i.e., neoskipping, aberrant splice-sites, and intron retentions were more frequent (Fig 2B). Although most of the splicing alterations did not affect the encoded ORF, neoskipping events impacted more frequently the ORF compared with the other event types (Fig 2B).

In total we found 2108 genes with predicted alterations in the protein product due to cancer cell specific splicing alterations, with a similar number of protein-affecting splicing changes in each cell line CA46: 1368, HL-60: 1043, and THP-I: 1700. Moreover, the predicted HLA-types from the RNA-seq data with PHLAT matched those previously reported [22]. We then predicted candidate MHC-I binding peptides (binding affinity $\leq$ 500nM) with NetMHC on all peptides, keeping only those splicing-derived peptides that were not encoded in the reference transcriptome. This produced 830 (CA46), 461 (HL-60), and 2072 (THP-1) candidate neoepitopes (Fig 2C and S2 Table). Neoskipping events produced more candidate neoepitopes in all cell lines compared with the other event types (Fig 2C). On the other hand, despite being less frequent, *de novo* exonizations produced a similar number of neoepitopes compared to intron retention events. Candidate self-epitopes that were affected by the splicing alteration were more common than the splicing-epitopes (Fig 2C). Moreover, separation of these candidate splicing-neoepitopes and self-epitopes according to HLA-types followed closely the results by cell line (Fig 2D), indicating an agreement in the MHC affinity of the peptides found in the cell lines and the HLA class predicted.

To test the potential therapeutic implications of these findings, we tested whether genes from two databases of treatment-associated responses were significantly represented in the set of genes with splicing-neoepitopes or splicing-affected self-epitopes. We observed genes linked

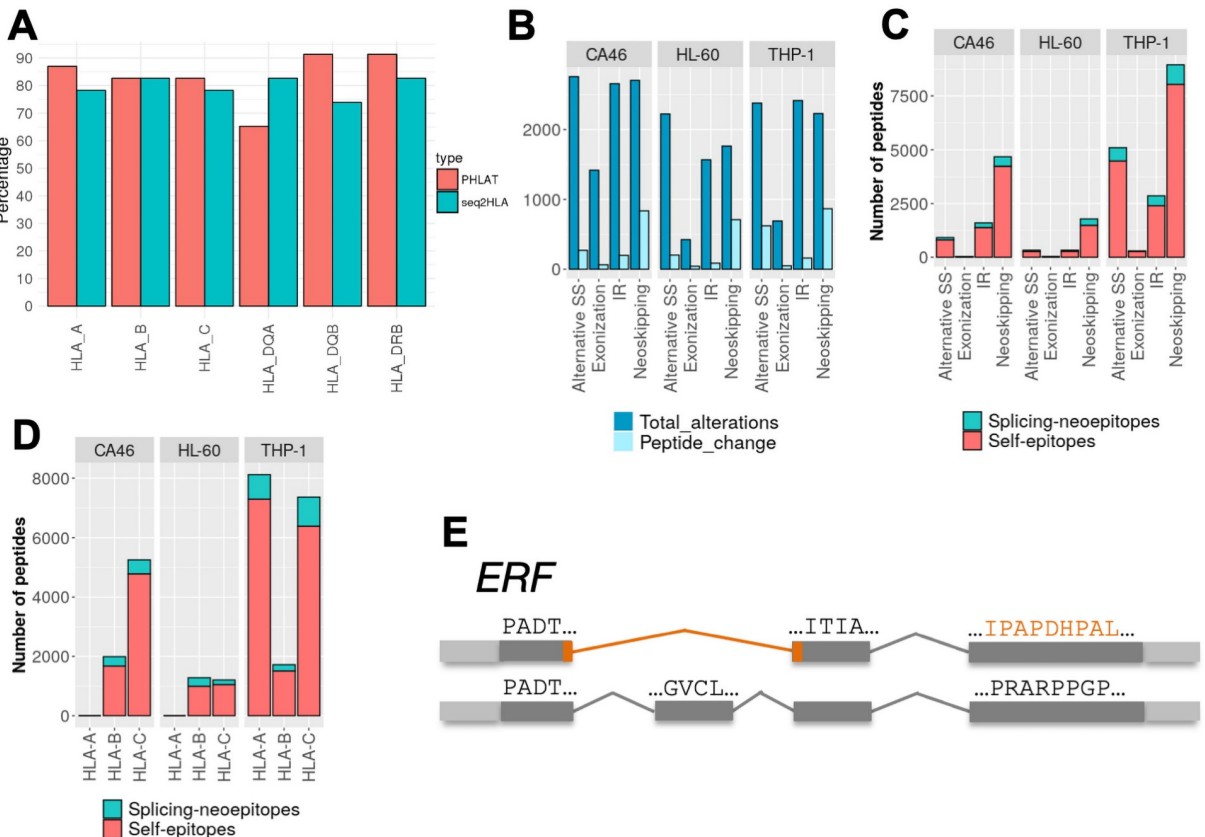

**Fig 2. Initial testing of ISOTOPE. (A)** Validation of the HLA type prediction from tumor RNA-seq data. We show the predictions for MHC Class I (HLA-A, HLA-B, HLA-C) and II (HLA-DQA, HLA-DQB, HLA-DRB) with PHLAT (red) and SeqHLA (blue). Each bar corresponds to the proportion of samples (over a total of 24 small cell lung cancer samples) for which the prediction on the tumor sample coincides with the prediction on the matched normal sample **(B)** For each cell line, CA46, HL-60 and THP-1, we show the number of different splicing alterations measured (dark blue) and the number of cases leading to a change in the encoded open reading frame (light blue). Alterations shown are alternative (5'/3') splice-site (A5_A3), *de novo* exonizations (Exonization), intron retentions (IR), and new exon skipping events (Neoskipping). **(C)** Number of splicing-derived neoepitopes (splicing-neoepitopes) (red) and splicing-affected self-epitopes (self-epitopes) (blue) detected for each of the splicing alterations in each of cell lines analyzed (CA46, HL-60 and THP-1). **(D)** as in (C) but separated by HLA-type. **(E)** Example of a splicing-neoepitope validated with MHC-I associated mass spectrometry data and derived from a neoskipping event in the gene *ERF*. The peptides are given in the same orientation as the 5' to 3' direction of the gene.

with therapy response in lymphoma were significantly represented in the set of self-epitopes (S3 Table). This result suggests a possible role of the splicing alterations detected in the involvement of these genes in therapy response. To validate the candidate splicing-derived neoepitopes we used MHC-1 associated mass-spectrometry (MS) data available for the same cell lines [22]. We identified three neoepitopes, all of them generated by neoskipping events in the genes *TOP1* (KRFEPLGMQK), *ERF* (IPAPDHPAL) and *IFRD2* (RTALGGMSW) (Fig 2E). These are different from the three peptides detected previously using the same datasets but only analyzing intron retention [16]. This disparity is possibly due to the different criteria used in the selection of relevant events. We performed an empirical test to keep only events with significant read support and considered other splicing alteration types beyond intron retention. Our results indicate that new types of splicing alteration can potentially produce tumor neoepitopes.

To further test ISOTOPE, we analyzed RNA-seq and MHC-I associated mass spectrometry data from ten breast cancer cell lines (MCF7, T47D, LY2, BT549, CAMA1, HCC1395,

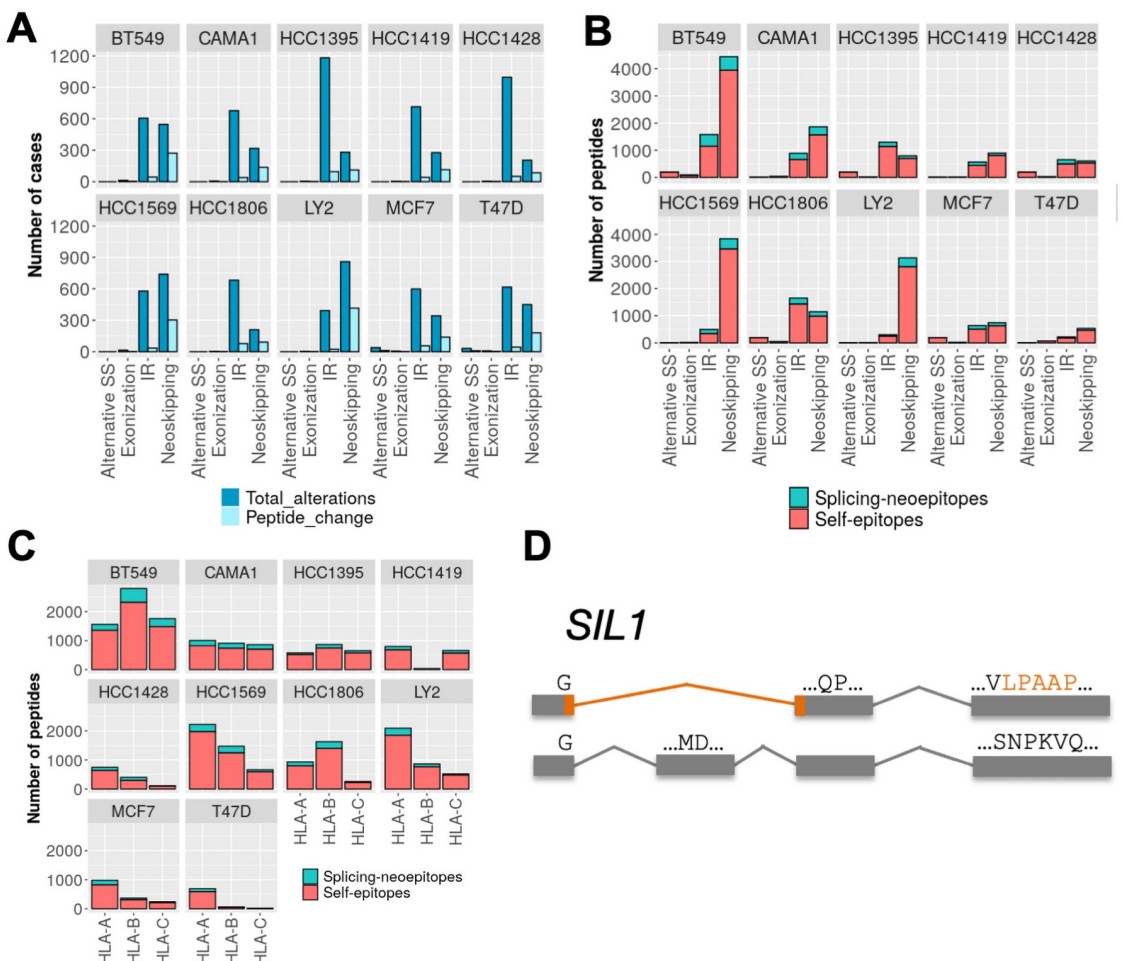

**Fig 3. Splicing epitopes in ten breast cancer cell lines. (A)** For each breast cancer cell line analyzed, the bar plots show the number of splicing alterations measured and the number of cases leading to a change in the reference protein. Alterations shown are alternative 5' or 3' splice-site (A5_A3), *de novo* exonizations (exonization), intron retentions (IR), and new exon skipping events (neoskipping). **(B)** Number of splicing-derived neoepitopes (splicing-neoepitopes) (red) and splicing-affected self-epitopes (self-epitopes) (blue) for each of the splicing alterations in each of the breast cancer cell lines tested. **(C)** as in (C) but separated by HLA-type. **(D)** Example of a splicing-derived neoepitope from a neoskipping event in the gene *SIL1* validated with MHC-I associated mass spectrometry in the cell line BT549.

HCC1419, HCC1428, HCC1569, HCC1806) [21,23]. The most frequent splicing alterations found were IR events, except for cell lines HCC1569 and LY2, for which neoskipping events were the most frequent (Fig 3A). However, for all types, neoskipping events produced the largest number of changes in ORFs in all cell lines. As before, we predicted the MHC-I binding potential for candidate epitopes, either splicing-derived neoepitopes or splicing-affected self-epitopes. Neoskipping events produced the largest number of neoepitopes (Fig 3B and S4 Table). As before, we observed more potentially affected self-epitopes than splicing-derived neoepitopes. Separating by HLA-type, splicing-derived neoepitopes were more frequently associated to HLA type A (Fig 3C). Additionally, we identified a significant association of genes involved in treatment response in breast cancer with genes producing splicing-derived neoepitopes (*ERBB2*, *ESR1*, *TIMP1*, *ABCC3*) or potentially depleted self-epitopes (*AKT1*, *CCNE1*, *RET*, *TFF3*). Next we searched the MHC-I associated mass spectrometry data for the same breast cancer cell lines [23] for the predicted neoepitopes. We only identified one

significant peptide match in the ten cell lines analyzed, which was generated by a neoskipping event resulting in a frameshift in the gene *SIL1* in BT549 (LPAAPLPLCPA, HLA-B) (Fig 3D).

## Tumor-specific splicing alterations impact self-epitopes and leads to more neoepitopes than mutations

We described above that tumor-specific splicing alterations potentially affect part of the open reading frame expressed in normal samples that could function as a self-epitope. To further investigate this, we analyzed a dataset of 123 small cell lung cancer (SCLC) patients [18,24–26]. SCLC is the most aggressive type of lung cancer, with a very early relapse after chemotherapy treatment and an average survival of 5% after 5 years of diagnosis [27]. SCLC is one of the cancer types with the largest TMB, which has been associated with its response to immune therapy [28]. Interestingly, SCLC presents a significantly higher density of mutations in introns compared to exons (Fig 4A), which may associate with a widespread impact on RNA-processing.

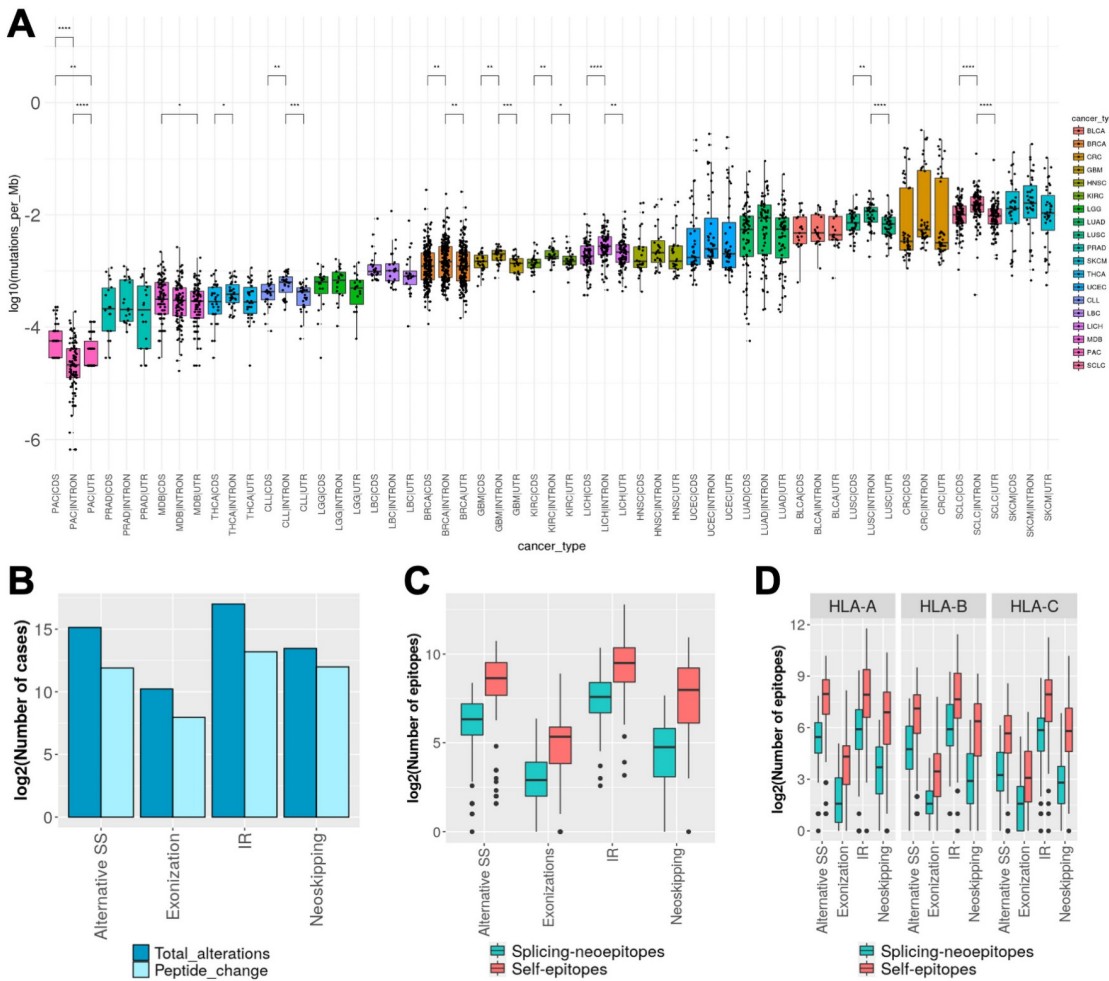

**Fig 4. Splicing epitopes in small cell lung cancer. (A)** Mutation burden (y axis) calculated separately for introns (INTRON), coding exons (CDS) and non-coding exonic regions in protein-coding genes (UTR) calculated from whole genome sequencing (WGS) data for several tumor types (x axis), including small cell lung cancer (SCLC). We indicate the pairs of distributions that were significantly different using a Wilcoxon test (* p-val <0.05, ** p-val <0.01, *** p-val<0.001, **** p-val<0.0001). **(B)** Number of splicing alterations (y axis) according to event type (x axis), indicating all alterations and the subset that impact the open reading frame (ORF). **(C)** Distribution of splicing-derived neoepitopes (splicing-neoepitopes) and splicing-affected self-epitopes (self-epitopes), separated by splicing alteration type. **(D)** Same as (C) but separated by HLA-type.

Accordingly, SCLC represents an interesting tumor type to investigate how splicing alterations may contribute to neoepitope burden in tumor cells.

We applied ISOTOPE to RNA-seq from 123 small cell lung cancer (SCLC) patients. We derived an exhaustive compendium of SCLC-specific splicing alterations by filtering out all SCLC junctions that appeared in a comprehensive dataset of normal splice junctions (S1 Fig). We found a total 14643 aberrant splice sites, 7039 intron retentions, 1311 neoskipping events, and 290 *de novo* exonizations that were SCLC specific, and were affecting 2955, 149, 620, and 169 genes, respectively (Fig 4B and S5 Table). The identified SCLC-specific splicing alterations distributed homogeneously across all samples and showed no association to mutations on spliceosomal factors or overexpression of MYC genes (S2 Fig). We focused on the SCLC-specific events that occurred within an ORF and could therefore alter the protein product: 3890 (27%) of the aberrant splice sites, 804 (61%) of the new skipping events, 753 (10%) of the intron retentions and 85 (29%) of the new exonizations (Fig 4B and S3 Fig).

To evaluate the immunogenic impacts induced by these splicing alterations, we predicted HLA-I and HLA-II types from the RNA-seq for the SCLC samples using PHLAT. We next used the altered and reference ORFs and searched for candidate MHC-I binders (binding affinity $\leq$ 500nM) that were specific to SCLC. We identified a total of 47,088 candidate splicing-derived neoepitopes, with the majority (60%) associated to intron retention events (S5 Table). On the other hand, we identified a total of 254,125 candidate splicing-affected self-epitopes (Fig 4C and 4D). This imbalance towards the potential elimination of self-epitopes occurred at the level of the number of predicted immunogenic peptides as well as the number of events producing immunogenic peptides. Moreover, this effect was not specific of any type of splicing alteration or HLA-type (Fig 4C and 4D).

We could not detect any significant association with SCLC-specific response biomarkers but did observe multiple significant associations of SCLC-specific splicing-derived neoepitopes with response biomarkers from other tissues (S3 Table). These results are especially relevant in SCLC, for which no alteration has been yet described as therapeutically targetable. We did not have access to MHC-I associated mass-spectrometry data for these SCLC samples. However, using mass-spectrometry data for MHC-I associated proteins in lymphoblasts [29] we were able to validate 1458 (11.7%) of the self-epitopes predicted to be potentially depleted in the altered isoform. To test the significance of the association of the predicted epitopes to the mass spectrometry data, we performed a randomized comparison. We took 1000 random peptides predicted with high affinity ($\leq$ 500nM) and 1000 peptides from the entire set of self-epitopes and checked how many from these 2 random sets would be validated by mass-spectrometry. We repeated this process 100 times and tested the difference of the two distributions. This analysis yielded a significantly higher number of validations for the candidate self-epitopes with high affinity (Kolmogorov-Smirnov p-value = 3.44e-13).

We further tested the association with tumor mutation burden (TMB). Overall, we found no association between the TMB and the number of splicing alterations or the number of epitopes (splicing-neoepitopes or self-epitopes). However, there was some association for the neoskipping events across all samples (S4 Fig). We next tested the association of splicing-derived neoepitopes with mutational neoepitopes. In the subset of SCLC samples with whole genome sequencing (WGS) data [24,25] there were more splicing-neoepitopes than mutational ones, and there was a weak correlation between their numbers in patients (Spearman rho = 0.397, p-value = 0.003) (S5 Fig). On the other hand, none of the splicing-neoepitopes matched any of the mutational neoepitopes.

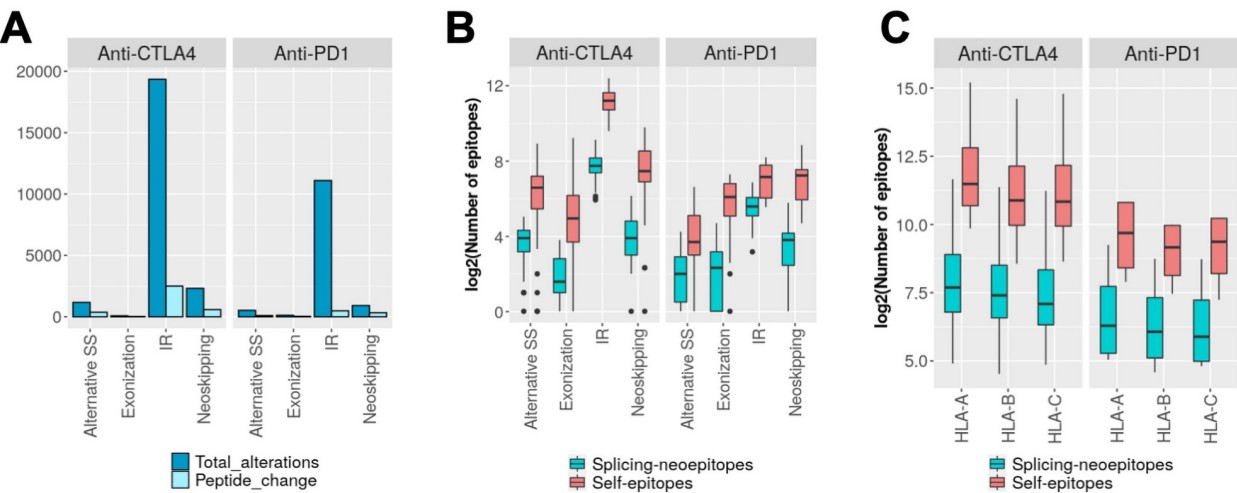

**Fig 5. Splicing epitopes in two melanoma cohorts. (A)** Total number of events and subset of protein-affecting events in the melanoma cohorts treated with anti-CTLA4 and with anti-PD1. **(B)** Distribution of the number of candidate tumor-specific splicing-derived neoepitopes (splicing-epitopes) and self-epitopes that would be depleted in the altered isoform (self-epitopes). **(C)** Distribution of the number of candidate epitopes from (B), separated by HLA-type.

## Association of splicing-derived epitopes with response to immune checkpoint inhibitors

To test whether tumor-specific splicing-derived neoepitopes may be associated to the patient response to immune therapy, we applied ISOTOPE to RNA-seq data from two cohorts of melanoma patient samples prior to treatment with anti-*CTLA4* [4] or anti-*PD1* [30] (S6 and S7 Tables). We calculated all the tumor-specific splicing alterations in each patient sample by removing all events that also occurred in a large set of control normal samples analyzed. Intron retention was the most abundant alteration but the impact on the encoded protein was not equally abundant in both cohorts (Fig 5A). Despite these differences, there was an overall decrease of the ORF lengths as a consequence of the splicing alterations (S6 Fig), in agreement with previous studies showing a reduction of ORF lengths expressed in tumors [31].

As for other samples tested, the overall number of splicing-affected self-epitopes was overall higher than the splicing-derived neoepitopes, with larger numbers of self-epitopes affected by intron retention events in the anti-*CTLA4* cohort (Fig 5B). Moreover, the anti-*CTLA4* cohort presented more epitopes from both classes for all HLA-types (Fig 5C). These results did not change when we used ≤300nM to define the candidate epitopes (S7 Fig). We compared the predicted neoepitopes in both sets with the annotated clinical response of the patient to the immunotherapy: responder or non-responder [4,30]. The number of splicing-derived neoepitopes in responders and non-responders in anti-*CTLA4* or anti-*PD1* showed no significant difference (S8 Fig). Separating the splicing alterations by type, we found in general a higher proportion of self-epitopes affected by splicing in all patients.

We did not observe any differences in the proportion of epitopes between responders and non-responders to anti-*CTLA4* (Fig 6A). However, responders to anti-*PD1* therapy had more splicing-affected self-epitopes from intron retention events compared to non-responders. Other splicing alterations did not show any significant differences (Fig 6B). We found similar results using the threshold 300nM to define candidates (S8 Fig). To further test the potential role of splicing-derived neoepitopes and splicing-affected self-epitopes in the response to

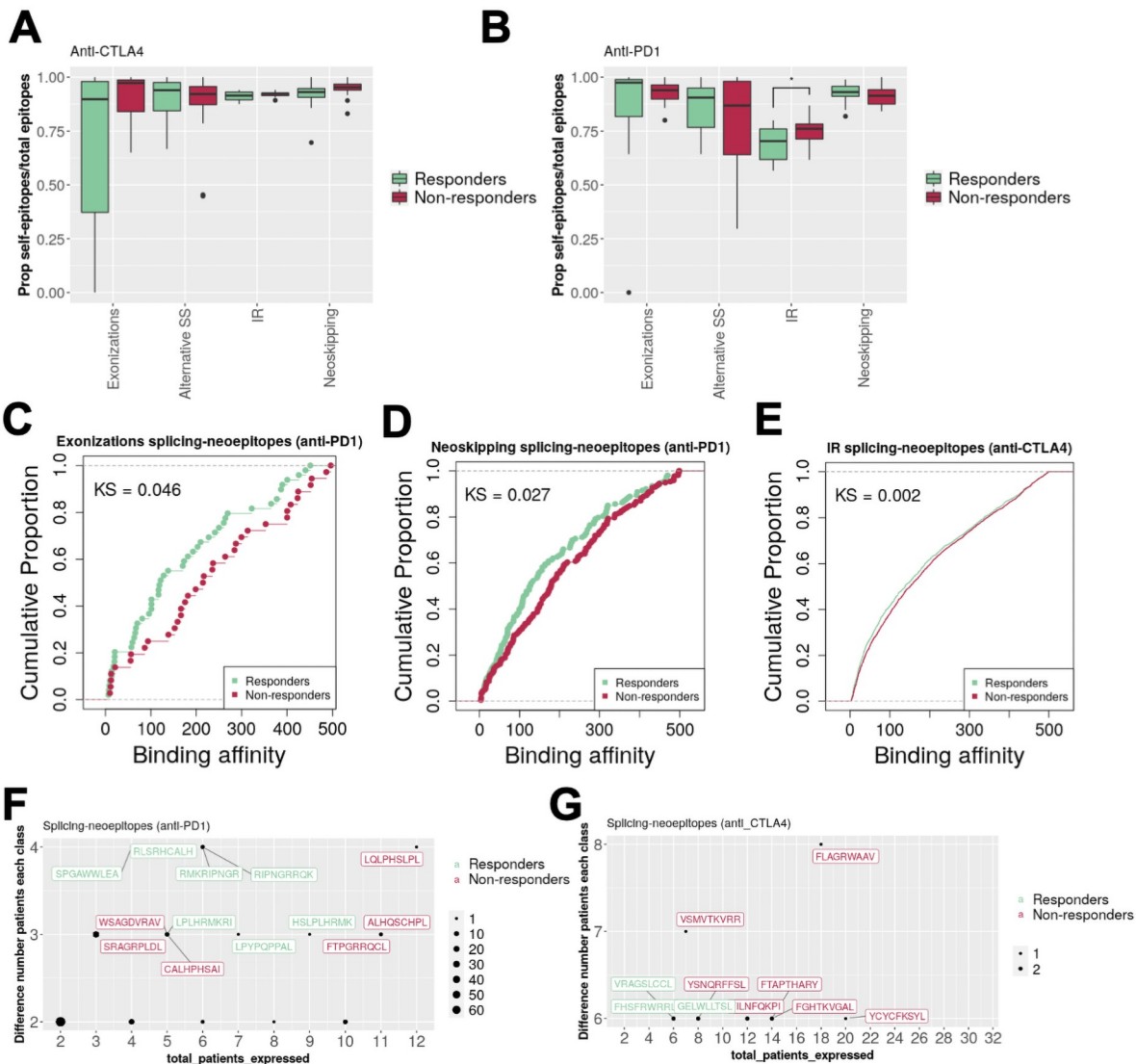

**Fig 6. Splicing epitopes and response to immune therapy. (A)** Proportion of splicing-affected self-epitopes (self-epitopes) over the total of epitopes, i.e., splicing-derived neoepitopes (splicing-neoepitopes) plus self-epitopes, (y axis) for patients treated with anti-*CTLA4*, separated by type of splicing alteration (x axis) and by patient response: responder (green) or non-responder (red). **(B)** As in (A) but for a different cohort of melanoma patients treated with anti-*PD1*. **(C)** Cumulative plots of the binding affinities (x axis) of splicing-neoepitopes in melanoma tumors from exonization events separated in responders (green) and non-responders (red) to anti-PD1 therapy. Kolmogorov-Smirnov test p-value (KS) = 0.0465 **(D)** As in (C), for splicing-derived neoepitopes from neoskipping events, KS = 0.0274. **(E)** Cumulative plots of the affinities of splicing-neoepitopes in melanoma tumors from intron retention events separated in responders (green) and non-responders (red) to anti-CTLA4 therapy, KS = 0.0016. **(F)** Frequency of splicing-neoepitopes represented according to the total number of patients in which are predicted (x axis, *total_patients_expressed*) and to the absolute count-difference in responders and non-responders to anti-*PD1* therapy (y axis, *Difference number patients each class*). Epitopes are indicated in green if they are more frequent in responders, and in red otherwise. The size of the point indicates the number of cases. **(G)** The same as in (F) but for responders and non-responders to anti-*CTLA4* therapy.

immunotherapy, we studied their binding affinities. Splicing-derived neoepitopes showed stronger MHC-I interaction (lower values of binding affinity) in responders. In particular, splicing-derived neoepitopes from *de novo* exonizations (Fig 6C) and neoskipping events (Fig 6D) had stronger interactions in anti-*PD1* responders, and those associated with intron-

retention events had stronger interactions in anti-*CTLA4* responders (Fig 6E). Incidentally, splicing-affected self-epitopes from exonizations in anti-*PD1* responders, and from new skipping events and intron retentions in anti-*CTLA4* responders also showed stronger MHC-I binding (S9 Fig).

When we considered the threshold of 300nM to define candidates, not all comparisons remained significant, but we observed similar trends (S10 Fig). We looked at various properties in the samples to test for possible confounding effects with the therapy response. The number of sequencing reads, and the overall distribution of transcript expression did not vary significantly between responders and non-responders (S11 Fig). We also observed no general association of response with the immune infiltration or purity of samples (S12 Fig). Moreover, although we observed a significant association between response and estimated stromal content in the anti-PD1 cohort, there was no correlation between the number of splicing-neoepitopes and the estimated stromal content (S12 Fig).

Finally, to further assess the potential relevance of the candidate splicing-derived epitopes, we studied whether the identified peptides occurred in multiple patients and in association with response to the immunotherapy. The most frequent splicing-derived neoepitope produced in responders for anti-*PD1* therapy was produced from an intron retention event in the proto-oncogene *PIM3* (SPGAWWLEA) and occurred in 4 out of 14 patients (29%), with HLA type HLA-B0702 (2 of them), HLA-B5501 and HLA-B5601 (Fig 6F). The most frequent splicing-derived neoepitopes in responders to anti-*CTLA4* therapy occurred in 6 out of 18 cases (33%) and were produced from intron retention events in the genes *SPTAN1* (FHSFRWRRL) and *GNAS* (VRAGSLCCL). All patients for both epitopes were of HLA type HLA-C0701 (Fig 6G).

## Discussion

Our comprehensive analysis cancer-specific splicing alterations indicate that splicing changes of any kind may potentially contribute to the immunopeptidome, hence they should be considered in studies of cancer and immunotherapy. Our approach, implemented in the pipeline ISOTOPE (https://github.com/comprna/ISOTOPE), presents several novelties and advantages with respect to previous approaches. It is exhaustive in the type of alterations tested, e.g., it includes *de novo* exonizations, which have not been previously characterized. Thus, making possible an assessment at unprecedented scale of candidate splicing-derived neoepitopes. Although tumor-associated intron retention is quite common in tumors [32], we observed that neoskipping events showed in greater proportion a disruption of the encoded proteins and led to more potential candidate neoepitopes. Moreover, unlike previous studies [14–16], our analysis describes tumor-specific alterations by comparing to a large compendium of normal samples and performs an empirical test to ensure that the cancer-specific splicing alteration considered is supported by significantly more reads than any other splicing alterations in the same locus. In our analyses we also tested potential MHC-II neo-epitopes. Although these predictions are generally less reliable, MHC-II associated neo-epitopes may also be relevant for immunotherapy [33,34]. Furthermore, ISOTOPE only requires RNA-seq data from a tumor sample, and it is applicable in the absence of DNA sequencing data from the tumor and without the need of RNA-seq data from a matched normal control. Additionally, unlike previous studies, we provide the software and instructions to run the complete ISOTOPE pipeline in a single computer or in a computer cluster. This makes possible its application on individual samples in a clinical setting, or on patient cohorts, similarly to the analyses presented. In summary, ISOTOPE enables a robust and exhaustive survey of the immunogenic impacts of splicing in cancer.

The low validation of splicing-neoepitopes with MHC-associated proteomics would suggest that there is a small contribution of tumor-specific splicing to novel epitopes, in agreement with previous studies. There are several possible reasons for that. The RNA-seq data used might have not been of sufficient depth to be able to robustly identify all relevant splicing alterations. This is suggested by the overall low recurrence of the tumor-specific splicing alterations found across patients. Although many of these might be accidental transcripts produced in a tumor, they still can change the identity of the tumor cell and shape their fitness. An additional reason may be related to the analysis of the proteomics data. MHC-I associated mass spectrometry does not use the enzymatic digestion standard in unbiased proteomics. Thus, to ensure that matches were reliably detected, we built a control dataset containing a large reference set of peptides, which could lead to a low detection rate. Additionally, we relied on candidate epitopes predicted from RNA-seq. However, a more sensitive approach might be based on the identification of splicing-derived neoepitopes directly from the MHC-I associated mass spectrometry. It is also possible that most peptides associated to the immune recognition of tumor samples may be produced through other mechanisms or may not be novel from the point of view of the expression pattern.

Recently, it was shown that for some tumor types, the occurrence of splicing alterations associates with higher expression of PD1 and PD-L1 [14]. It was then suggested that these tumors could benefit from immune checkpoint inhibitor therapy due to the presence of a higher content of splicing-derived neoepitopes. However, in a recent study of neoepitopes derived from intron retention [16], no association was found between neoepitope count and the response to checkpoint inhibitors. Here, we extended this comparison to all other types of splicing alterations. Using two cohorts of patients treated with immune checkpoint inhibitors we found no differences in the number of splicing-derived neoepitopes between responders and non-responders. However, we observed differences in the predicted affinity to the MHC-I complex. Indeed, the overall interaction strength predicted for neoepitopes in responders was larger, possibly indicating a better recognition of tumor cells in the immune response triggered by the treatment. This raises the possibility that splicing-derived neoepitopes may contribute to the positive response to the therapy. On the other hand, our analysis indicated a weak correlation of the number of splicing-neoepitopes with tumor mutation burden, which has been previously shown to correlate with immune therapy response. But splicing-neoepitopes were generally more abundant than mutational neoepitopes and showed no overlap between them. This suggests that splicing-neoepitopes may represent biomarkers of immunogenicity independently of the mutational patterns. Further analyses in different cancer types will be needed to further explore this exciting possibility.

We have also studied the possibility that splicing alterations could affect the open reading frame in such a way that certain self-epitopes are no longer produced. This raises the interesting question about the impacts that the lack of these self-epitopes might have. Although T-cell selection in the thymus can remove some of these self-reactive specificities, it is known that this could be incomplete or suboptimal [35]. Self-peptides might not bind equally well to MHC molecules, which would then compromise the efficiency of negative selection of self-reactive T-cells. As a consequence, potentially self-reactive T-cells can be found in circulation in healthy individuals [36], and tolerance to some self-antigens could often rely on the additional control of regulatory T-cells expressing *CTLA4* [37]. This suggests an intriguing possibility. Upon treatment with immune-checkpoint inhibitors, among the immunocompetent T-cells that are freed to act against the tumor cells, there may be some with self-reactive capabilities. These may help destroying tumor cells but could also be a potential trigger of autoimmune responses. Indeed, treatment with immune-checkpoint inhibitors has led to serious and sometimes fatal autoimmune reactions in patients [38–40]. T-cells may attack the tumor cells via

neoepitopes as well as self-epitopes but could trigger immune responses through the reactivity against self-epitopes in normal cells. On the other hand, a depletion of self-epitopes may lead to a reduced response. We have observed that some melanoma patients with self-epitope depletion show no response to the treatment. Also, when we characterized the splicing alterations in small-cell lung cancer, a tumor type with low survival and with limited response to immune therapy, self-epitope depletion occurs much more frequently than splicing-neoepitope production, and we could validate many of them from MHC-associated mass spectrometry in lymphocytes. Thus, tumor-specific splicing alterations could generate neoepitopes, but could also potentially deplete self-epitopes. These alterations may not necessarily prevent the self-reactivity in normal cells but could reduce the recognition and destruction of tumor cells, thereby hindering the effect of the immune therapy. This suggests the interesting hypothesis that tumor-specific splicing alterations may contribute to the escape of tumor cells to immune-checkpoint inhibitor treatment.

As the ability of the immune system to identify malignant cells relies on the tumor cells maintaining sufficient antigenicity, it is thus essential to exhaustively explore all potential immunogenic impacts, including the variety of splicing alterations that may arise in tumors. Our method ISOTOPE facilitates this exploration in individual samples and in patient cohorts, thereby helping in the identification of molecular markers of response to immunotherapy.

## Methods

### Datasets

RNA sequencing (RNA-seq) data for the cell lines analyzed was collected from the cancer cell line encyclopedia (CCLE) [21] (GEO accession number GSE36139). We also collected RNA-seq data from 38 melanoma patients pre anti-CTLA4 treatment classified as responder (18 cases) and non-responder (20 cases) [4], available from dbGAP (https://www.ncbi.nlm.nih.gov/gap) under accession phs000452.v2.p1: and RNA-seq data from 27 melanoma patients pre anti-PD1 treatment [30], available at SRA (https://www.ncbi.nlm.nih.gov/sra) under accession SRP070710, also classified as responder (14 cases) or non-responder (13 cases). Additionally, we gathered RNA-seq data from 123 SCLC patients [24] (EGA accession EGAS00001000925), [18] (EGA accession EGAD00001000223), and [26]. For the SCLC patients from [18] we also obtained the matched normal controls. Samples with more than 30% of junctions present in other samples but with missing value in them, were filtered out. We estimated the stromal content, immune infiltrate, and tumor purity of every sample from gene expression information using the ESTIMATE R package (v.1.0.13) [41].

### Identification of tumor-specific splicing alterations

All RNA-seq samples were mapped to the genome (hg19) using STAR [42] and were processed as described before [43]: Mapped spliced reads with at least a common splice site across two or more samples were clustered with LeafCutter [44], with a minimum of 30 reads per cluster and a minimum fraction of reads of 0.01 in a cluster supporting a junction. Read counts per junction were normalized over the total of reads in a cluster. Junction clusters were defined across all patients but normalized read counts were calculated per patient. Junctions were classified as novel if either or both of the splice-sites were not present in the human annotation (Gencode v19) [45], they had at least 10 supporting reads in at least one tumor sample, and did not appear in any of the normal samples from a comprehensive dataset collected from multiple sources: 7859 normal samples from 18 tissues from the GTEX consortium [46], normal samples from Intropolis [47], CHESS 2.0 [48], and 24 matched normal samples from lung [18].

ISOTOPE classifies the novel junctions in clusters as one of the following types: aberrant splice-site, new exon skipping (neoskipping), or *de novo* exonization. To define exonizations, we considered all pairs of spliced junctions that were not present in normal samples (see above) that would define a potential new internal exon not longer than 500nt, with flanking canonical splice site motifs (AG-GT). We kept only cases with more than 5 reads validating each splice site. For tumor specific neoskippings, we considered those new junctions that skipped known exons and defined new connections between exons. To define retained introns (RIs) we used KMA [49] to extend the Gencode (v19) transcriptome with potential retained introns (RIs), which we quantified in each RNA-seq sample with Kallisto [50]. To filter out RIs that were not tumor specific, we calculated RI events with SUPPA [51] from the human Gencode [45] and the CHESS 2.0 [48] annotations, and removed KMA-predicted RIs that appeared in the SUPPA RI annotations. To control for confounding effects due to defects in pre-mRNA processing across the entire gene locus, for each splicing alteration we compared the expression of the alterations with 100 randomly selected cases from the same gene using an Empirical Cumulative Distribution Function (ECDF) test. Candidate junctions were compared with other junctions, exonizations were compared with genic regions of similar length, and retained introns were compared with other introns. Cell line data was processed in a similar way, but without removing the alterations in normal samples, as those tests were focused on the presentation of splicing-derived neo-epitopes.

## Protein impact of the splicing alterations

For each analyzed cohort, we built a reference transcriptome using the largest mean expression per gene across samples, using only those cases with mean > 1 transcript per million (TPM). Transcript abundance was calculated using Salmon [52]. A reference proteome was defined from these reference transcripts. For each splicing alteration, a modified transcript was then built using as scaffold the reference transcript exon-intron structure. Unless the splicing alteration only affected the untranslated region (UTR), an altered protein was calculated from the longest open reading frame (ORF) (start to stop) predicted on the modified transcript. Each splicing alteration was considered only if an ORF was predicted. If the splicing alteration deleted the region of the start codon, the closest downstream start codon was used. Further, if the stop codon in the altered ORF was located further than 50nt from a downstream splice site, the case was discarded as potential NMD target. Software to run this analysis and selection of novel splicing junctions is available at http://github.com/comprna/ISOTOPE.

## Prediction of splicing-derived neoepitopes

ISOTOPE calculates two types of epitopes. One type corresponds to tumor-specific splicing-derived neoepitopes. These are peptides with affinity to the MHC-I complex that are not encoded in the wild-type transcripts but are encoded in the altered ORF as a consequence of the tumor-specific splicing alteration. The second type corresponds to splicing-affected self-epitopes. These are peptides with affinity to the MHC-I complex that are encoded in the wild-type transcripts but would not be encoded in the altered ORF, i.e. potentially depleted, as a consequence of the tumor-specific splicing alteration.

Unless available, we inferred the HLA-type from the tumor RNA-seq using PHLAT [19]. From all proteins derived from the splicing alterations and from the reference proteome, we predicted potential MHC-I binders with NetMHC-4.0.0 [17], and with NetMHCpan-4.0 [53] for the classes missing in NetMHC-4.0.0. Those peptides in common between the reference and the altered protein were discarded. Peptides in the altered protein with binding affinity $\leq$ 500nM, but not present in the reference proteome were considered candidate

neo-epitopes. We performed the same analysis for MHC-II binders using predictions from NetMHCII-2.3, and complementing them with the predictions from NetMHCIIpan-3.2 for the missing types [54].

### Validation of neoepitope prediction with MHC-I mass-spectrometry

MHC-I associated mass-spectrometry data was analyzed following the approach from [55]. We tested all the candidate neoepitopes using as a control database all candidate MHC-I binders from Uniprot. Using as a control all candidate MHC-I binders from the reference proteome yielded similar results. Using candidate binders rather than the entire Uniprot reduces the search space and takes into account that MHC-I proteomics involves unspecific digestion. We matched the mass spectra to the joined set of control and candidate splicing neoepitopes, and with the control set alone. Candidate matches for both sets were compared to calculate their significance. For the analysis of the MHC-I associated data for the cell lines CA46, HL-60 and THP-I, we used the same procedures as described before for these datasets [16]. To test the significance of the identification of the predicted epitopes in the mass-spectrometry data from [29], a randomized comparison was performed. We took two random sets of 1000 random predicted epitopes each, one set with cases of good affinity ($\leq$ 500nM) and one set from all the set of predicted neoepitopes (with or without good affinity). We then checked how many of these 2 random sets are validated with mass-spectrometry data from [29]. We repeated this process 100 times and tested with a Kolmogorov-Smirnov test whether the 2 distributions of the number of peptides validated were significantly different. For the self-epitopes in SCLC was significant (p-value = 3.44e-13), whereas for splicing-neoepitopes there was no significant difference.

### Somatic mutation data and detection of mutation-derived neoepitopes

We used somatic mutations from whole genome sequencing for 505 tumor samples from 14 tumor types [56]: bladder carcinoma (BLCA) (21 samples), breast carcinoma (BRCA) (96 samples), colorectal carcinoma (CRC) (42 samples), glioblastoma multiforme (GBM) (27 samples), head and neck squamous carcinoma (HNSC) (27 samples), kidney chromophobe (KICH) (15 samples), kidney renal carcinoma (KIRC) (29 samples), low grade glioma (LGG) (18 samples), lung adenocarcinoma (LUAD) (46 samples), lung squamous cell carcinoma (LUSC) (45 samples), prostate adenocarcinoma (PRAD) (20 samples), skin carcinoma (SKCM) (38 samples), thyroid carcinoma (THCA) (34 samples), and uterine corpus endometrial carcinoma (UCEC) (47 samples). Additionally, we used whole-genome somatic mutation calls for SCLC from [24] (EGA accession EGAS00001000925). We only used substitutions and discarded those that overlapped with frequent (>1% allele frequency) SNPs (dbSNP 144).

Mutation-derived epitopes were calculated with pVACtools [57], using whole genome sequencing data (WGS) for two SCLC cohorts [24,25]. The identification of splicing-derived neoepitopes was carried out with ISOTOPE using the RNA-seq data from the same patient samples. The candidate epitopes were calculated in both cases using the same tools, NetMHC and NetMHCPan, with the same parameters, testing peptides with amino acid length from 8 to 11, and selecting candidates with binding affinities less or equal than 500nM.

### Biomarker enrichment analysis

The Clinical Interpretation of Variants in Cancer (CIViC) [58] and the Cancer Genome Interpreter (CGI) [59] databases were used to identify biomarkers, in the form of genetic alterations, associated to the treatment response to anti-cancer therapy. Genes with predicted

splicing-neoepitopes or self-epitopes in each cohort were assessed for enrichment in biomarkers by means of a Fisher Test and multiple test correction (FDR estimation).

## Supporting information

**S1 Fig. Properties of the SCLC samples.** Purity analysis of the small cell lung cancer (SCLC) samples from each one of the three cohorts used for this study: George et al. [24] (**A**), Iwakawa et al. [26] (**B**), and Rudin et al. [18] (**C**). In each case, we give the distribution of tumor purity values (between 0 and 1) calculated with ESTIMATE [41]. Length distributions of the new exons produced as a consequence of aberrant splice sites (**D**) or new exonizations (**E**). The lengths follow extreme value distributions with mean values of 100, similar to known exons. (PDF)

**S2 Fig. SCLC-specific splicing alterations.** From top to bottom, the number of mapped spliced reads, the expression of the MYC genes (known to be amplified or overexpressed in SCLC and to drive splicing alterations), mutations on core spliceosome factors, tumor mutation burden and the number of the different event types detected by ISOTOPE for the SCLC samples (blue from George et al. [24], red from Rudin et al. [18], green from Iwakawa et al. [26]). (PDF)

**S3 Fig. Splicing-derived epitopes and splicing-affected self-epitopes in SCLC patients.** (**A**) Upper panel: Number of intron retentions per SCLC sample that impact the open reading frame. Lower panel: Number of candidate MHC-I binders per sample that are created (blue), i.e., splicing-derived neoepitopes, or potentially removed from the ORF by the splicing alteration (red) through exonizations. (**B**) Same as in (A) but for neoskipping events. (PDF)

**S4 Fig. Correlation of events and neoepitopes with the tumor mutation burden.** (**A**) Correlations between the number of splicing alterations detected and the tumor mutation burden (TMB) for all the SCLC patients, separated by splicing alteration type. Although across all the events types the correlation is low (Spearman $\rho = 0.182$), separately there was a statistically significant correlation for Neoskipping events ($\rho = 0.42$). We show the same correlations separating splicing-derived neoepitopes (**B**) and splicing-affected self-epitopes. (**C**). Although neoskipping events showed significant association, there was an overall low correlation across all the event types between the TMB and the splicing-neoepitopes ($\rho = 0.194$) and self-epitopes ($\rho = 0.196$). (PDF)

**S5 Fig. Comparison between splicing-derived neoepitopes and neoepitopes derived from somatic mutations.** (**A**) Comparison of the number of neoepitopes derived from somatic mutations (red) and tumor-specific splicing-derived neoepitopes (blue) in the SCLC patient cohorts from Peifer et al. [25] and from George et al. [24] (**B**) For each patient from the same cohorts, we give the number of neoepitopes derived from somatic mutations (x axis) and the number of neoepitopes derived from tumor-specific splicing alterations (y axis). Mutation-derived epitopes were calculated with pVACtools, whereas splicing-derived neoepitopes were calculated with ISOTOPE as described in the manuscript. The candidate epitopes were calculated in both cases using the same tools with the same parameters: NetMHC and NetMHCPan, using the hg19 reference, testing peptides with amino acid length from 8 to 11, and selecting candidates with binding affinities less or equal than 500nM. There were no overlaps between candidates generated by both methods. (PDF)

**S6 Fig. Length differences between the wild type (WT) open reading frame (ORF) and the splicing altered ORF.** We show the distributions of the length ratios between WT ORF and the ORF affected by the splicing alteration for the anti-PD1 **(A)** and the anti-CTLA4 **(B)** cohort. The ratios are plotted in log2 scale, i.e., log2(WT length/aberrant length). The plots are separated according to whether the change involved the creation of a splicing-derived neoepitope only (blue), the removal of a splicing-affected self-epitope only (green), or both (red). We plot in the lower panels the proportion of the total corresponding to each case.
(PDF)

**S7 Fig. Splicing-associated epitopes identified using ≤300nM. (A)** Distribution of the number of candidate tumor-specific splicing-derived neoepitopes (splicing-epitopes) and splicing-affected self-epitopes that would be depleted in the altered isoform (self-epitopes) using ≤300nM to define candidate epitopes. **(B)** Distribution of the number of candidate epitopes from (A), separated by HLA-type.
(PDF)

**S8 Fig. Splicing-associated epitopes and immune therapy response. (A)** Distribution of the number of tumor-specific splicing-derived neoepitopes (splicing-epitopes) and splicing-affected self-epitopes that would be depleted in the altered isoform (self-epitopes) using ≤500nM to define candidate epitopes, separated by clinical outcome. The number of splicing-derived neoepitopes in responders to anti-CTLA4 (mean 346) and non-responders (mean 375) were not significantly different. Similarly, the anti-PD1 cohort showed no significant difference between the total number of splicing-derived neoepitopes between responders (mean 46) and non-responders (mean 62.6) in the anti-PD1 cohort. **(B)** as in (A) but using ≤300nM to define candidates. The number of tumor-specific splicing-derived epitopes in responders to anti-CTLA4 (median 172) and non-responders (median 228) were not significantly different. A similar result was found for the self-epitopes (2090 and 2159). We found the same for the anti-PD1 cohort (splicing tumor-epitopes: 51.5 vs 79; splicing self-epitopes: 269 vs 287). **(C)** Proportion of splicing-affected self-epitopes over the total of epitopes (splicing-affected self-epitopes and tumor-specific splicing-derived neoepitopes) (y axis) for patients treated with anti-*CTLA4*, separated by type of splicing alteration (x axis) and by patient response: responder (green) or non-responder (red). In this plot, candidate epitopes were defined using ≤300nM as threshold. **(D)** As in (B) but for melanoma patients treated with anti-*PD1*.
(PDF)

**S9 Fig. Analysis of the epitope affinities in responders and non-responders using ≤500nM to define candidates. (A)** Cumulative plot of the binding affinities (x axis) of splicing-affected self-epitopes in melanoma tumors from exonization events separated in responders (green) and non-responders (red) to anti-PD1 therapy. Smaller values of binding affinity correspond to a stronger interaction between the peptides and the MHC-I complex. We also give the Kolmogorov-Smirnov test p-value (KS) = 0.0074. **(B)** Cumulative plot of the binding affinities (x axis) of splicing-affected self-epitopes in melanoma tumors from new skipping events (neoskipping) events separated in responders (green) and non-responders (red) to anti-*CTLA4* therapy, KS = 0. **(C)** Cumulative plots of the affinities of splicing-affected self-epitopes in melanoma tumors from intron retention events separated in responders (green) and non-responders (red) to anti-*CTLA4* therapy, KS = 0.
(PDF)

**S10 Fig. Analysis of the epitope affinities in responders and non-responders using ≤300nM to define candidates. (A)** Cumulative plot of the binding affinities (x axis) of exonization-derived neoepitopes in melanoma tumors separated in responders (green) and

non-responders (red) to anti-*PD1* therapy. Smaller values of binding affinity correspond to a stronger interaction between the peptides and the MHC-I complex. We also give the Kolmogorov-Smirnov test p-value (KS). **(B)** Cumulative plot of the binding affinities (x axis) of neoskipping-derived neoepitopes in melanoma tumors from separated in responders (green) and non-responders (red) to anti-*PD1* therapy. **(C)** Cumulative plots of the affinities of intron-retention-derived neoepitopes in melanoma tumors separated in responders (green) and non-responders (red) to anti-*CTLA4* therapy.
(PDF)

**S11 Fig. Comparison of sample properties between responders and non-responders.** We show the number of mapped reads in responder and non-responder patients in the anti-CTLA4 **(A)** and the anti-PD1 **(B)** cohorts. We also show the distribution of the transcript expression values for each patient, represented as log2(TPM) (y axis) for the anti-*CTLA4* **(C)** and for the anti-*PD1* **(D)** cohorts.
(PDF)

**S12 Fig. Stroma and Immune content comparisons between responders and non-responders.** We show the stromal content (StromalScore), immune cell infiltration (ImmuneScore), and overall score predicted with ESTIMATE separating patients according to the treatment response in each cohort, anti-*PD1* **(A)** and anti-*CTLA4* **(B)**. The only significant differences detected was in relation to the stromal content in the anti-PD1 cohort (p-value ~ 0.05). **(C)** Number of splicing neo-epitopes (y axis) as a function of the stromal score (x axis).
(PDF)

**S1 Table. HLA predictions for Rudin et al. [18] RNA-seq samples.**
(XLSX)

**S2 Table. Epitopes predicted from cell lines studied in Smart et al, 2018 [16].**
(XLSX)

**S3 Table. Biomarkers enrichment analysis.** Disease and therapeutic association, i.e., combination of biomarker, drug, evidence level and response. FDR: False Discovery Rate.
(TXT)

**S4 Table. Epitopes predicted from breast cancer cell lines (Rozanov et al, 2018 [23]).**
(XLSX)

**S5 Table. Epitopes predicted from SCLC samples.** Epitopes predicted from SCLC samples (George et al, 2015 [24], Rudin et al, 2012[18]).
(XLSX)

**S6 Table. Epitopes predicted from anti-PD1 samples (Hugo et al, 2016 [30]).**
(XLSX)

**S7 Table. Epitopes predicted in the samples from the anti-CTLA4 cohort.** Epitopes predicted from anti-CTLA4 samples (Van Allen et al, 2015 [4]).
(XLSX)

## Acknowledgments

We thank the authors from the different studies used in this article for facilitating access to their RNA-seq datasets: Iwakawa et al. [26], Van Allen et al [4] (phs000452.v2.p1), Hugo et al. [30] (SRP070710), George et al. [24] (EGAS00001000925), Rudin et al. [18] (EGAD00001000223).

## Author Contributions

**Conceptualization:** Eduardo Eyras.

**Data curation:** Juan L. Trincado, Marina Reixachs-Solé, Judith Pérez-Granado, Tim Fugmann, Jun Yokota.

**Formal analysis:** Juan L. Trincado, Marina Reixachs-Solé, Judith Pérez-Granado, Tim Fugmann.

**Investigation:** Jun Yokota.

**Methodology:** Juan L. Trincado, Eduardo Eyras.

**Project administration:** Eduardo Eyras.

**Resources:** Jun Yokota.

**Software:** Juan L. Trincado.

**Supervision:** Ferran Sanz, Jun Yokota, Eduardo Eyras.

**Visualization:** Juan L. Trincado.

**Writing – original draft:** Juan L. Trincado, Eduardo Eyras.

**Writing – review & editing:** Juan L. Trincado, Ferran Sanz, Eduardo Eyras.

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
