## [Decision Letter · Decision Letter 0]

9 Dec 2020

Dear Dr. Eyras,

Thank you very much for submitting your manuscript "ISOTOPE: ISOform-guided prediction of epiTOPEs in cancer" for consideration at PLOS Computational Biology.

As with all papers reviewed by the journal, your manuscript was reviewed by members of the editorial board and by several independent reviewers. In light of the reviews (below this email), we would like to invite the resubmission of a significantly-revised version that takes into account the reviewers' comments.

We cannot make any decision about publication until we have seen the revised manuscript and your response to the reviewers' comments. Your revised manuscript is also likely to be sent to reviewers for further evaluation.

In melanoma it is known that in some cases immune response is targeted against self-antigens in genes involved in pigmentations, for example PMEL (SILV), MLANA, and such immune response is associated with loss, of variable degree, of skin pigmentation. There is also some association between degree of autoimmune side effects and therapeutic benefits from immune check point inhibitors. However, as indicated by reviewers there are number of questions which need to be carefully addressed regarding possibility of loss of self-antigens due to splicing and putting such potential mechanism in context of other known mechanisms of immune escape.

ISOTOPE method as indicated by reviewers has potential to generate useful analysis, however there are number of questions raised by reviewers which need to be carefully addressed. One of areas which needs to be addressed is providing much more details on method, including important relevant aspects of existing tools used by this method, for example exon junction reads in context of exon skipping analysis. Also, including information potential method users likely would want to be included, for example information on transcripts MANE Select/Plus status, gene names, etc.

Sincerely,

Dmitriy Sonkin, PhD

Guest Editor

PLOS Computational Biology

Florian Markowetz

Deputy Editor

PLOS Computational Biology

Reviewer's Responses to Questions

**Comments to the Authors:**

Reviewer #1: Manuscript Review: ISOTOPE: ISOform-guided prediction of epiTOPES in cancer

Summary

The authors present a method, “ISOTOPE” for identifying splice isoform variants in RNA-seq samples derived from cancer patients. The authors present data on the validation of this method, and then apply it to large genomic data sets to identify patterns of different splice isoforms in cancer. A recurrent theme is how splice isoforms lead to depletion of “self antigens”, which the authors reference numerous times.

In general, while the method itself is potentially interesting, the analysis of results generated using this method is problematic with a range of issues and questions detailed below. Claims around the depletion of self antigens are confusing and it is unclear if there is a particular significance of this and also, what the appropriate comparison group is. I have included my notes below - as it stands there are so many issues with this manuscript I have only listed the most pressing ones here.

Concerns:

Intro: Does this method require a special/certain type of sequencing? How can it identify *all* cancer-specific splicing alterations? That is a strong claim.

The description of the method in the main text is highly perfunctory and confusing. Would suggest including more detail here and also referencing the methods more heavily.

Method validation: it would seem to be important for the authors’ approach that they can in fact identify bonafide neoepitopes with their method. They claim in their validation section that “We were able to detect three neoepitopes on three different samples” - this feels rather weak to me, given that 100s or 1000s of neoepitopes are being predicted. For any method, I would expect some to hit even by chance. The authors should do a more thorough analysis demonstrating that their approach can indeed predict and prioritize true tumor presented epitopes better than previous and/or naive approaches.

“Gained and depleted enrichment analysis”: the claim there are genes “enriched” for depleted self-antigens (ATM, EZH2, REL1) is a very strong one, since this would imply recurrent immune selection against a presumably wildtype peptide. If this is true substantially more evidence, included as figures and statistical analyses, is required. In general, the presentation of anecdotal evidence for a subset of genes like above is not sufficient for a claim, since it is unknown how many total genes were tested in this setting.

Neoantigen prediction:

“Peptides in the altered protein with binding affinity ≤ 500nM, but not present in the reference proteome were considered candidate neo-epitopes.”: 500 nM is a very weak cutoff and much stronger cutoffs have been recently proposed [cell]. Please show your analysis and conclusions are not dependent on this value.

Furthermore, expression level is known to be a key determinant of tumor epitope immunogenicity. Since your calls are made from RNA-seq, I would strongly recommend including an expression estimate for your candidate splice-isoform derived neoepitopes. Notably, since these can co-exist with the wildtype isoform, it will be important to establish how the expression of these epitopes can be assessed in an isoform-specific manner.

Splice isoforms lead to depletion of candidate epitopes: The authors make a big claim around the fact that splice isoforms can lead to depletion of other “candidate epitopes”, however, their methodology and claims are unclear. Are the “candidate epitopes” all epitopes (wildtype +mutated)? If so, this claim is not surprising, since it is comparing only non-cancer epitopes from splice isoforms to any epitopes in the native isoform. A more interesting comparison would be to compare splice isoform derived neoepitopes to somatic mutation-derived neoepitopes in the same sample - to what extent to splice isoforms subsume somatic mutation-derived neoepitopes?

Figure 5: This figure, with multiple panels, is only referenced in passing. It is standard to require a reference to each panel in a figure for it to be included, and as is I do not believe this figure adds much to this manuscript as written. The authors should either eliminate this figure or include more writing describing the results in it.

Analysis of data from immune checkpoint blockade treated patients.

Panels 6a and b: In each panel, the authors test a range of hypotheses and identify a single statistically significant one. The approach is problematic in that typically, a multiple hypothesis correction would be required. Furthermore, it appears that the proportion of epitopes lost in both healthy and control for the PD1/IR group is less than in CTLA4, even though these samples are both at pretreatment. So I would hypothesize there is potentially a batch effect present somewhere in this analysis. The authors should perform a more careful analysis, controlling for multiple hypothesis testing as well as potentially correcting batch effects in their data to ensure this result is correct.

Panels 6c, d, e: In general I find this analysis unconvincing, since it counts the clinical data from each patient multiple times (comparing R vs. NR, but among all epitopes, i.e, N=total number of epitopes, whereas the R vs. NR is among total number of patients). It is important in any analysis comparing clinical variables that each patient be counted only once (i.e., that N=the total number of patients). The authors should fix the issues with this analysis and perform an analysis with the appropriate comparator groups.

Reviewer #2: 1. Software and manual.

I have downloaded the ISOTOPE. However, the authors wrote that "the scripts are ready to be run in a slurm cluster.", and I do not have access to one. It will be good to provide the version of the pipeline for stand-alone linux computers. Also, in the manual the authors wrote that "Until all jobs generated by a part are finished do not run the following part.". It would be useful to provide a sequential script for this purposes. Overall, the manual should be more user-friendly and detailed.

2. Statistical tests. The authors write that "ISOTOPE performs a bootstrapping test to establish the significance of the read support of the candidate splicing alterations. This type of test has not been performed in previous similar analyses of splicing-derived neoepitopes and ensures the robustness of the events detected". It is important to explain the choice of bootstrapping method and why bootstrapping is better than other statistical methods to assess the accuracy of estimates.

3. Structure of the manuscript. Some of the paragraphs in the results section may be more appropriate for the Methods. For example, discussion of the selection of method between PHLAT (Bai et al. 2014) and Seq2HLA (Boegel et al. 2018) may be more suitable for the Methods, or a Supplement.

4. Comparison with other available methods and tools has not been done.

Reviewer #3: Trincado et al

ISOTOPE: ISOform-guided prediction of epiTOPES in cancer

This paper describes the creation of a novel bioinformatic pipeline to predict the change in MHC class I epitope profile in a tumor specimen using whole transcriptome RNAseq data. The pipeline is developed using cell line data, and its functionality is replicated on other cell lines and on tumor samples. Most interestingly, the ISOTOPE method predicts that many more self-epitopes are lost in tumor samples than neoepitopes are gained. The authors hypothesize that the loss of self-epitopes may be an important driver of immunogenicity in cancer, and that self-epitope loss may be a biomarker for response to immunotherapy.

While the observation that self-epitope loss is common in tumors is a provocative and interesting finding, it is not well-established what biological or clinical relevance this may have from this study. Analysis of a melanoma cohort is provided that does show that for one class of splicing alteration in one therapy type there is a small difference in proportion of immunotherapy responders and non-responders. However, it is hard to judge the significance of this finding in the absence of more clinical data or a better understanding of the underlying biology.

Major Comments:

1. This paper would be strengthened significantly by better tying the phenomenon of self-epitope loss to a clinical or biological consequence. As the authors observe, the melanoma cohort used here is likely not ideal for analysis of this phenomenon, since mutation burden is very high in this tumor and the effects of epitope gain/loss due to splicing are likely diluted by mutation-generated neoepitopes. Analysis in another tumor type might be more fruitful. Admittedly, it is very difficult to find datasets like this to test the pipeline on, but showing a clear significance of the phenomenon would make it a lot more convincing.

2. This study does take some steps to look at the relationship between Tumor Mutation Burden (TMB) and epitope gain/loss. Because TMB is the best understood measure of immunogenicity, it would be really helpful to get a sense of how TMB and epitope gain/loss are related. It is stated that for the SCLC dataset, TMB and the number of epitope gains/losses is not correlated based on Supplemental Figure 2. This does appear to be true, but a more rigorous treatment would be helpful, as this is an important point – what is numerically, the relationship between TMB and neoepitope gain, epitope loss, total epitope changes, etc.

Further, there is a clear difference between the three cohorts used in this analysis – for epitope gains/losses, clearly Georg>Rudin>Iwakawa. Do the correlations/lack of correlations hold up within each cohort?

I find it surprising that number of epitopes lost/gained does NOT correlate to TMB, since you’d think the same underlying process would be causative for both (i.e. DNA mutations). This is exciting, because a mutational signature that is not correlated to TMB may reflect some different underlying biology that independently affects immunogenicity, and which may be a powerful biomarker in combination with TMB.

3. Analysis of all datasets is broken down by splicing mutation type. It would be helpful to see the overall data for each dataset with all mutation types lumped together, if there is no biological reason to think that a neoepitope or an epitope loss generated by each type of splicing event is different. The simplest model to test is that the overall level of epitope loss correlates to a clinical outcome, and it would be helpful to try to develop a single metric for this that could be used as a biomarker like TMB.

Minor comments:

-In figure 5A the colors for total peptides and peptides changed seem to have been reversed?

-Does the number of self-antigens lost correlate to the quantity of coding sequence lost? I can see how skipping an exon can remove an antigen, but it is less obvious how epitopes are lost in other mechanisms that add sequence (like intron retention). Are these all losses of epitopes that span splice junctions? Is it a result of sequence loss that accompanies gains?

-Can the authors comment on the relative importance of the tumor/normal comparison and the use of splicing data from outside non-tumor datasets to define the set of “normal” splice junctions? Is it possible to devise a pipeline that doesn’t need a comparison to a normal patient sample for tumor-only sequencing, which is common in the clinical setting?

-In figure 6b the proportions of epitopes lost are much less for both responders and non-responders are much less (~75% vs ~95%) for intron retentions than for all other event categories shown and for IR in the CTLA4-treated group. Can the authors comment on why this may be? Also, is this significant to why this was the only segment of the data that showed a difference between responders and non-responders?

- In support of Figure 6 it is stated that the number of splicing-generated neoepitopes is not different between responders and non-responders. I don’t think these data are shown, so this should be at least cited as “data not shown”, though I think it’s an interesting analysis that should be included.

- Please make data labels bigger. I had a really hard time reading some of them. Also “A5_A3” isn’t intuitive at all and should be replaced with “alternative splice site” as in figure 1

**Have all data underlying the figures and results presented in the manuscript been provided?**

Reviewer #1: Yes

Reviewer #2: Yes

Reviewer #3: None

PLOS authors have the option to publish the peer review history of their article (what does this mean?). If published, this will include your full peer review and any attached files.

Reviewer #1: No

Reviewer #2: No

Reviewer #3: No
---

## [Decision Letter · Decision Letter 1]

4 Jun 2021

Dear Professor Eyras,

Thank you very much for submitting your manuscript "ISOTOPE: ISOform-guided prediction of epiTOPEs in cancer" for consideration at PLOS Computational Biology. As with all papers reviewed by the journal, your manuscript was reviewed by members of the editorial board and by several independent reviewers. The reviewers appreciated the attention to an important topic. Based on the reviews, we are likely to accept this manuscript for publication, providing that you modify the manuscript according to the review recommendations.

Comments from reviewer 2 need to be fully addressed. In particular, version of the pipeline for stand-alone Linux computers to run on small test dataset need to be provided, this is important for providing convenient way for reviewers and readers to test and experiment with pipeline on small data sets. Supplemental tables value is currently limited by absence of gene symbols, this prevents readers from quickly and conveniently searching for isoform(s) in gene of interest.

Sincerely,

Dmitriy Sonkin, PhD

Guest Editor

PLOS Computational Biology

Florian Markowetz

Deputy Editor

PLOS Computational Biology

[LINK]

Comments from reviewer 2 need to be fully addressed. In particular, version of the pipeline for stand-alone Linux computers to run on small test dataset need to be provided, this is important for providing convenient way for reviewers and readers to test and experiment with pipeline on small data sets. Supplemental tables value is currently limited by absence of gene symbols, this prevents readers from quickly and conveniently searching for isoform(s) in gene of interest.

Reviewer's Responses to Questions

**Comments to the Authors:**

Reviewer #2: The manual remains to be not user-friendly. The hyperlinks were the test dataset did not work. Without the sample dataset it is not possible to evaluate the pipeline. Good user-friendly program needs to have a built-in dataset(s) and several well-described usage examples.

Regarding the comparison software, Part I has been implemented by many other tools (https://journals.plos.org/plosone/article?id=10.1371/journal.pone.0156132, https://bioconductor.org/packages/release/bioc/vignettes/SGSeq/inst/doc/SGSeq.html, https://arxiv.org/pdf/1405.0788.pdf). It is important to assess the quality of exon/intron predictions.

Reviewer #3: My thanks to the authors for their thorough response to the reviewers' comments. My review is concerned mainly with the clinical utility and biological insight gained from this new method of analysis.

My major comment was that the study should attempt to better explain how this novel method for analysis of splicing alterations in tumors could be used clinically or provide biological insight. While the authors have not convincingly demonstrated an immediate use for their method, I think they have done what's possible with public data, and that further investigation is probably a new study and outside of the scope of this report. My comments have been appropriately answered on all other counts.

**Have the authors made all data and (if applicable) computational code underlying the findings in their manuscript fully available?**

Reviewer #2: Yes

Reviewer #3: None

PLOS authors have the option to publish the peer review history of their article (what does this mean?). If published, this will include your full peer review and any attached files.

Reviewer #2: No

Reviewer #3: No

Figure Files:

Data Requirements:

Reproducibility:

References:

---

## [Decision Letter · Decision Letter 2]

30 Aug 2021

Dear Professor Eyras,

We are pleased to inform you that your manuscript 'ISOTOPE: ISOform-guided prediction of epiTOPEs in cancer' has been provisionally accepted for publication in PLOS Computational Biology.

Best regards,

Dmitriy Sonkin, PhD

Guest Editor

PLOS Computational Biology

Florian Markowetz

Deputy Editor

PLOS Computational Biology

Reviewer's Responses to Questions

**Comments to the Authors:**

Reviewer #2: The authors have made all required improvement. The software is publicly available and now the manual is user-friendly.

**Have the authors made all data and (if applicable) computational code underlying the findings in their manuscript fully available?**

Reviewer #2: Yes

PLOS authors have the option to publish the peer review history of their article (what does this mean?). If published, this will include your full peer review and any attached files.

Reviewer #2: No

---

## [Editor Report · Acceptance letter]

10 Sep 2021

PCOMPBIOL-D-20-01709R2 

ISOTOPE: ISOform-guided prediction of epiTOPEs in cancer

Dear Dr Eyras,

I am pleased to inform you that your manuscript has been formally accepted for publication in PLOS Computational Biology. Your manuscript is now with our production department and you will be notified of the publication date in due course.

With kind regards,

Olena Szabo
